

# Hydrogen Peroxide Photoformation in Particulate Matter and its Contribution to S(IV) Oxidation During Winter in Fairbanks, Alaska

Michael Oluwatoyin Sunday[1*], Laura Marie Dahler Heinlein[1], Junwei He[1], Allison Moon[2], Sukriti Kapur[3], Ting Fang[3**], Kasey C. Edwards[3], Fangzhou Guo[4***], Jack Dibb[5], James H. Flynn III[4], Becky Alexander[2], Manabu Shiraiwa[3], Cort Anastasio[1]

[1]Department of Land, Air and Water Resources, University of California, Davis, California 95616, USA
[2]Department of Atmospheric and Climate Science, University of Washington, Seattle, Washington, 98195, USA
[3]Department of Chemistry, University of California, Irvine, California 92697, USA
[4]Department of Earth & Atmospheric Sciences, University of Houston, Houston, Texas 77204, USA
[5]Earth Systems Research Center, University of New Hampshire, Durham, New Hampshire 03824 USA

* Now at: California Air Resources Board, Sacramento, CA, USA
** Now at: Sustainable Energy and Environment Thrust, The Hong Kong University of Science and Technology (Guangzhou), Guangzhou, Guangdong 511400, China
*** Now at: Center for Atmospheric and Environmental Chemistry, Aerodyne Research Inc, Billerica, Massachusetts 01821, USA

Correspondence to: Cort Anastasio (canastasio@ucdavis.edu); Michael Sunday (mosunday@ucdavis.edu)

## Abstract

The high levels of sulfate in wintertime particles in Fairbanks, Alaska are a subject of keen research interest and regulatory concern. Recent results from the 2022 ALPACA (Alaska Layered Pollution And Chemical Analysis) field campaign indicate that roughly 40% of wintertime sulfate in Fairbanks is secondary, with hydrogen peroxide (HOOH) the dominant oxidant. Since formation of HOOH in the gas phase should be negligible during ALPACA because of high levels of $NO_x$, we examined whether reactions within particles could be a significant source of HOOH. To test this, we collected particulate matter (PM) samples during the ALPACA campaign, extracted them, illuminated them with simulated sunlight, and measured HOOH production. Aqueous extracts showed significant light absorption, a result of brown carbon (BrC) from sources such as residential wood combustion. Photoformation rates of HOOH in the PM extracts (normalized to Fairbanks winter sunlight) range from 6 to 71 µM hr$^{-1}$. While light absorption is nearly independent of pH, HOOH formation rates decrease with increasing pH. Extrapolating to the concentrated conditions of aerosol liquid water (ALW) gives an average rate of in-particle HOOH formation of ~ 0.1 M hr$^{-1}$. Corresponding rates of sulfate formation from particle-produced HOOH are 0.05 – 0.5 µg m$^{-3}$ hr$^{-1}$, accounting for a significant portion of the secondary sulfate production rate. Our results show that HOOH formed in





particles is an important contribution to sulfate formation in ambient wintertime particles, even under the low actinic flux conditions typical of winter in subarctic locations like Fairbanks.

## 1 Introduction

Levels of fine particulate matter (PM$_{2.5}$) in Fairbanks, Alaska during winter frequently exceed the U.S. Environmental Protection Agency (U.S. EPA) 24-hour limit of 35 μg m$^{-3}$ (ADEC, 2023). During pollution events, the dry particle mass is composed of roughly 60 – 80% organic compounds, 20% sulfate, and smaller amounts of other inorganic ions (Nattinger, 2016; Philip et al., 2014; Snider et al., 2016; Wang and Hopke, 2014; Ward et al., 2012). The Alaskan Layered Pollution And Chemical Analysis (ALPACA) field campaign in Fairbanks during January – February of 2022 (Simpson et al., 2024) showed that most sulfate is directly emitted, but that secondary sulfate formation accounts for roughly 40% of particulate sulfate (Moon et al., 2024). Based on isotopic measurements of particulate sulfate, the dominant oxidant forming secondary sulfate was hydrogen peroxide (HOOH), with smaller contributions from ozone, nitrogen dioxide, and other pathways (Moon et al., 2024).

Aqueous HOOH reacts rapidly with dissolved sulfur dioxide (i.e., inorganic S(IV)) (Hoffmann and Edwards, 1975) and this reaction is an important source of sulfuric acid/sulfate in cloud/fog drops and particle water (Seinfeld & Pandis, 2016; Song et al., 2021a; Ye et al., 2021). Typically, hydrogen peroxide is formed in the gas phase, via combination of hydroperoxyl radicals (HO$_2$•), and then partitions into the atmospheric aqueous phase where it oxidizes inorganic S(IV) (Atkinson et al., 2004; Jackson and Hewitt, 1999; Reeves and Penkett, 2003; Xuan et al., 2020; Ye et al., 2018). But under high-NO$_x$ conditions – such as in Fairbanks during winter (EPA, 2020) – little gas-phase HOOH should be formed because HO$_2$• concentrations are suppressed by NO (Jackson & Hewitt, 1999; Lee et al., 2000; Ye et al., 2021). However, hydrogen peroxide may still be an important oxidant for inorganic S(IV) under these conditions because it can be photochemically formed in particles (Song et al., 2023; Ye et al., 2021).

There is a long history of HOOH photoformation measurements in environmental samples. This includes both surface waters such as rivers and marine waters (Cooper et al., 1988; Olasehinde et al., 2008; Shaked et al., 2010; Sunday et al., 2020) as well as cloud/fog drops and aerosol particles (Anastasio et al., 1994; Arakaki et al., 1995; Cooper et al., 1988; Faust et al., 1993; Shaked et al., 2010; Song et al., 2023; Ye et al., 2021; Zuo and Hoigné, 1993). There are multiple chromophores (i.e., light-absorbing species) and mechanisms that can form HOOH in atmospheric drops and particles. Zuo & Hoigné (1993) reported that illumination of Fe(III)-oxalate complexes forms Fe(II) and HOOH. Similarly, Ye et al. (2021) reported that complexes of transition metal ions (TMIs) and humic-like substances in particles generate HOOH. However, metals are not required to photochemically form hydrogen peroxide: Anastasio et al. (1997) showed that triplet states of aromatic carbonyl photosensitizers in the presence of phenols efficiently form HOOH in aqueous solution. Since both aromatic carbonyls and phenols are major emissions from residential wood combustion (Schauer et al., 2001), which is widespread in Fairbanks during





winter (Nattinger, 2016; Simpson et al., 2024; Wang and Hopke, 2014), this mechanism might be a significant path for HOOH formation in during ALPACA.

While it is typically thought that photochemistry is negligible at high latitudes in winter, we hypothesize that biomass-burning brown carbon (BrC) in wintertime particles from Fairbanks, Alaska can photochemically form HOOH. Recent studies by Kaur et al. (2019) and Ma et al. (2024) on illuminated extracts of wintertime particles in Davis, California reported the formation of photooxidants, including singlet molecular oxygen ($^1O_2*$) and oxidizing triplet excited states of organic matter ($^3C*$), from the photoexcitation of BrC. Similarly, BrC and other chromophores in wintertime particles from Fairbanks, Alaska absorb sunlight

and form $^3C*$, $^1O_2*$, and hydroxyl radical ($^\bullet OH$) (Heinlein et al., In preparation). These studies demonstrate that BrC-mediated photochemical reactions of particulate chromophores form oxidants, although they did not measure HOOH.

To address this gap, here we examine whether aqueous extracts of winter particles from Fairbanks can form HOOH. We report the first measurements of HOOH formation in wintertime particles from a high-latitude site and describe the potential role of HOOH in oxidizing dissolved $SO_2$ to particulate sulfate. Our objectives are to: (1) measure the formation and loss of HOOH

in aqueous extracts of ALPACA PM samples, (2) extrapolate from our dilute extract results to concentrated ambient aerosol liquid water (ALW) conditions, and (3) estimate the rate of sulfate formation by HOOH in ALW and assess its contribution to the sulfate content of the PM samples.

## 2 Experimental

### 2.1 Chemicals

All chemicals were used as received. Hydrogen peroxide ($\geq 30\%$), ethylenediamine tetraacetic acid disodium dihydrate (ACS reagent) and horseradish peroxidase were from Sigma-Aldrich. Potassium hydrogen phthalate ($> 99\%$) and sulfuric acid (trace metal grade) were from Fisher Scientific. 4-hydroxyphenyl acetic acid was from TCI chemicals. Ammonium hydroxide (28 – 30%) was from ACROS Organics. All chemical solutions were prepared using air-saturated purified water (Milli-Q water) from a Milli-Q Advantage A10 system (Millipore; $\geq 18.2$ M$\Omega$ cm).

### 2.2 Particle sampling and extraction

Particulate matter was sampled from two sites: the House site (64.850°N, 147.676°W), which was in a residential area in the Shannon Park neighborhood of Northeast Fairbanks, and the CTC site (64.841°N, 147.727°W), which was in downtown Fairbanks. At the House site, a Hi-Vol sampler (Tisch Environmental; flow rate $\approx 1.1$ m$^3$ min$^{-1}$) collected 24-hour PM$_{2.5}$ samples on prebaked 8 inch × 10 inch micro quartz filters (Whatman QMA). At the CTC site, a high-volume sampler (Tisch

Environmental, TE-5170; flow rate $\approx 1.6$ m$^3$ min$^{-1}$) equipped with a 4-stage cascade impactor (TE-230) was used to collect size-segregated 24-hour PM samples onto prebaked quartz filters; our experiments used the TEQMA filters from the final stage of the impactor, which collected particles with diameters $\leq 0.7$ μm. Details of sampling are in Edwards et al. (2024) and



Moon et al. (2024) for the House and CTC sites, respectively. After shipping to UC Irvine (House samples) or U Washington (CTC samples), a quarter (9 cm × 11 cm) of each filter was cut off, wrapped in pre-baked aluminum foil, sealed in a Ziplock

bag and transported in coolers with ice packs to UC Davis, where they were stored at -20 °C until processing. To obtain sufficient PM mass for illumination experiments, we grouped 2 - 13 consecutively collected filters into composites (Table S1). We refer to each composite by the middle day in the sampling period, formatted as dd/mm.

To prepare our PM extracts, we cut several 2 cm × 2 cm squares from each filter in the composite. Each square was placed in a separate 20 mL amber vial and extracted with 1.0 mL of 0.05 M (pH 1) sulfuric acid by mechanical shaking for 4 hours. The

same number of squares were extracted from each filter for a given composite. pH 1 was our standard extract acidity to mimic the low-pH mode of ambient particles in Fairbanks during the campaign (Campbell et al., 2024). We also extracted some samples in pH 3 or 5 sulfuric acid to examine the impact of acidity. Extracts from a given composite were pooled together and filtered using a 0.2 µm syringe filter (Pall Scientific). For each composite, a separate set of squares was also extracted in Milli-Q water to determine the PM-mass/water-mass ratio, dissolved organic carbon (DOC), and ion composition. The pH and UV-

Visible absorption spectra of the extracts were immediately measured after filtration using a pH microelectrode (MI-414, Microelectrodes, Inc.) and a Shimadzu UV-2501PC spectrophotometer, respectively. Extracts were then divided into 3 mL HDPE bottles, flash-frozen in liquid nitrogen, and stored at -20 °C until further analysis.

We investigated the effect of particle dilution on HOOH photoformation by preparing multiple dilutions of the same composite. This series was made by extracting squares from the 2/14 CTC composite filters with five different volumes of pH 1 sulfuric

acid per filter square: 10.0, 2.0, 0.70, 0.40 and 0.30 mL (Ma et al., 2023). For the two most concentrated extracts (0.40 and 0.30 mL of solvent per square filter), we first extracted multiple squares with 1.0 mL of 0.020 M and 0.015 M sulfuric acid, respectively. The individual extracts from each dilution were pooled together, filtered using a 0.2 µm syringe filter, then concentrated to the equivalent of 0.40 or 0.30 mL solvent per square using a rotary evaporator (Buchi Rotavapor R110) kept at 65 - 70 °C with a recirculating water bath (Isotemp 4100, Fischer Scientific). The concentration factors (CF) for the dilution

series of extractions, defined as the inverse of the volume of extract solvent used, were 0.10, 0.50, 1.43, 2.50 and 3.33.

**2.3 Light absorption properties**

We determined light absorbance by each filtered PM extract (PME) immediately after preparation in a 1 cm quartz cuvette. The absorbance ($A_\lambda$) was converted to a base-10 light absorption coefficient ($\alpha_\lambda$, cm$^{-1}$) using:

$$\alpha_\lambda = \frac{A_\lambda}{l}, \qquad (1)$$

where $l$ is the path length in cm. The rate of sunlight absorption in each PM extract under Fairbanks, Alaska, winter actinic flux ($R_{abs,PME,AK}$, mol-photons L$^{-1}$ s$^{-1}$) was calculated as





$$R_{\text{abs,PME,AK}} = 2.303 \times \frac{10^3}{N_{\text{A}}} \times \sum_{310 \text{ nm}}^{550 \text{ nm}} (\alpha_\lambda \times I_{\lambda,\text{AK}} \times \Delta\lambda) \times 2.5 , \qquad (2)$$

where 2.303 is for base conversion, $10^3$ is for unit conversion (cm$^3$ to L), $N_{\text{A}}$ is Avogadro's number, $I_{\lambda,\text{AK}}$ is the Fairbanks midday, winter-solstice actinic flux (photons cm$^{-2}$ s$^{-1}$ nm$^{-1}$) measured during the composite period, and $\Delta\lambda$ is the wavelength

interval (1 nm) between actinic flux data points. The actinic flux for a given composite is the average of the three peak hourly values (midday, one hour prior, and one hour after) averaged across each day included in the composite. While we measured the downwelling actinic flux, this was corrected to include the upwelling component based on several measurements of albedo during the campaign (Heinlein et al., In preparation). The factor of 2.5 at the end of equation 2 represents the estimated enhancement in actinic flux in suspended particles due to optical confinement (Corral Arroyo et al., 2022).

We determined the rate of light absorption by extracts in our experiments during illumination with simulated sunlight ($R_{\text{abs,PME,EXP}}$) using an equation analogous to equation 2 with two alterations: the photon flux ($I_{\lambda,\text{EXP}}$) in the illuminated sample was determined using 2-nitrobenzaldehyde actinometry (Hullar et al., 2020) and normalized to a standard $j_{2\text{NB}}$ value of 0.020 s$^{-1}$ (see section 2.4), and the factor of 2.5 was not applied.

### 2.4. Illumination and HOOH quantification

We illuminated our thawed extracts with a solar simulator consisting of a 1000 W Xenon arc lamp (XBO 1000W/HS, OSRAM) and three downstream optical filters: a water filter, an AM1.0 air mass filter (AM1D-3L, Sciencetech), and a 295 nm long-pass filter (20CGA-295, Thorlabs). The temperature of the sample chamber was maintained at 10 $^\circ$C using a recirculating water bath (Isotemp 3013D, Fisher Scientific).

We thawed a frozen extract in the dark at room temperature then transferred 1.0 mL to a cylindrical GE 021 quartz tube (5 mm

internal diameter, 1 mL volume) stoppered with a silicon plug. The entire solution was irradiated with simulated sunlight without stirring. At specific time intervals, a 200 µL aliquot of the irradiated extract was withdrawn for HOOH quantification using HPLC with the post-column derivatization and fluorescence detection method of Kok et al. (1995), as described previously (Chu and Anastasio, 2005). A stock solution of HOOH was freshly prepared in Milli-Q on each day of analysis and its concentration was determined on a UV-Visible spectrophotometer using a molar absorptivity of 38.1 M$^{-1}$ cm$^{-1}$ at 240 nm

(Miller and Kester, 1988). Calibration standards from the stock were prepared fresh in Milli-Q water on each experiment day.

The experimentally measured HOOH photoformation rate ($P^*_{\text{HOOH,PME,EXP}}$) and first-order-loss rate constant of HOOH ($k_{\text{HOOH,PME,EXP}}$) in the PM extracts were determined by fitting measurements of [HOOH] as a function of illumination time using an exponential rise to maximum equation:

$$[\text{HOOH}]_t = [\text{HOOH}]_0 e^{-k_{\text{HOOH,PME,EXP}} \times t} + \frac{P^*_{\text{HOOH,PME,EXP}}}{k_{\text{HOOH,PME,EXP}}} \left(1 - e^{-k_{\text{HOOH,PME,EXP}} \times t}\right), \qquad (3)$$



where $[HOOH]_t$ and $[HOOH]_0$ are the concentrations of HOOH at times $t$ and zero, respectively. On every experiment day, we measured the photon flux in the solar simulator using 2-nitrobenzaldehyde actinometry (Galbavy et al., 2010). We used this value to normalize each laboratory HOOH photoformation rate to a constant $j_{2NB}$ value of 0.020 s$^{-1}$ (a typical value from our experiments) to correct for variations in light intensity on different days (Table S3); we label the $j_{2NB}$-normalized rate as $P_{HOOH,PME,EXP}$.

We determined the quantum yield of HOOH formation ($\Phi_{HOOH}$) in each extract during solar simulator illumination using:

$$\Phi_{HOOH} = {P_{HOOH,PME,EXP}} \Big/ {R_{abs,PME,EXP}} \; . \qquad (4)$$

For each composite, we calculated $P_{HOOH,PME,AK}$, the photoformation rate of HOOH expected in the extract under Fairbanks midday sunlight for that composite, as $\Phi_{HOOH} \times R_{abs,PME,AK}$.

**2.5 Particle composition**

We measured water-soluble organic carbon in our extract composites using a TOC analyzer (Shimadzu TOC-VCPH), major inorganic anions ($SO_4^{2-}$, $NO_3^-$, $NO_2^-$, $Cl^-$, $Br^-$, and $PO_4^{3-}$) and cations ($NH_4^+$, $Na^+$, $K^+$, $Mg^{2+}$, and $Ca^{2+}$) with ion chromatography (Dionex ICS-6000), and metals (Fe, Cu, Mn, V, and Ni) using Inductively-Coupled Plasma – Mass Spectrometry (8900 triple quadrupole, Agilent technologies). For anion analysis we used a Dionex$^{TM}$ IonPac$^{TM}$ AS18-Fast-4μm column with an eluent of 10 mM KOH, while for cations we used a Dionex$^{TM}$ IonPac$^{TM}$ CS12 column with 20 mM

methanesulfonic acid as eluent.





# 3 Results and discussion

## 3.1 Extract characteristics and light absorption properties

The PM mass/liquid-water mass ratios of the extracts are in the range $(1.7 - 4.7) \times 10^{-4}$ µg PM/µg $H_2O$ (Table S1), similar to
values for extracts of wintertime particles from Davis, CA (Kaur et al., 2019). These ratios are typical of cloud and fog drops
and indicate that our extracts are roughly $10^3$ - $10^4$ times more dilute than ambient particles, which have a ratio near 1 µg
PM/µg $H_2O$ (Kaur et al., 2019). The major inorganic ions in the extracts are ammonium ($NH_4^+$, 291 – 2442 µM), sulfate ($SO_4^{2-}$, 187 – 1058 µM), and nitrate ($NO_3^-$, 84 – 701 µM) (Table S2), while dissolved organic carbon (DOC) concentrations range
from 1670 to 4910 µmol-C $L^{-1}$ (i.e., 20 – 59 mg-C $L^{-1}$; Table S1).

Figure 1 shows absorption spectra for composite extracts collected near the beginning and end of the ALPACA campaign,
along with the actinic flux for each composite. The absorbance of our extracts declines exponentially with increasing
wavelength, with average ($\pm 1\sigma$) absorption coefficients of $0.41 \pm 0.21$ and $0.11 \pm 0.05$ $cm^{-1}$ at 300 and 365 nm, respectively
(Table S1). The bulk of this absorbance is likely from brown carbon (BrC) with additional contributions from metal-organic
complexes and other chromophores. Values of $\alpha_{300}$ for our samples (0.20 – 0.98 $cm^{-1}$) are similar to values for Davis CA
wintertime PM extracts ($0.44 \pm 0.16$ $cm^{-1}$) (Kaur et al., 2019). The average DOC-normalized mass absorption coefficients (i.e.,
$MAC_{DOC}$ values) of our extracts at 300 and 365 nm are $2.89 \pm 0.80$ and $0.78 \pm 0.23$ $m^2$ $(g\text{-}C)^{-1}$, respectively (Table S1). These
are similar to values for wintertime particle extracts from Davis CA (Kaur et al., 2019; Ma et al., 2024), wintertime particle
extracts from China (Du et al., 2014; Yu et al., 2023; Zhang et al., 2024), and water extracts of biomass-burning particles (Fan
et al., 2018; Park and Yu, 2016). The Angstrom absorption exponents (AAEs) of our extracts range from 8.2 - 10.2 for House
particles and 7.1 – 8.2 for PM samples from the CTC site (Table S1). These values are in the range of typical values reported
for aqueous extracts of particles from biomass burning (Hecobian et al., 2010; Hoffer et al., 2006; Kaur et al., 2019; Ma et al.,
2023).






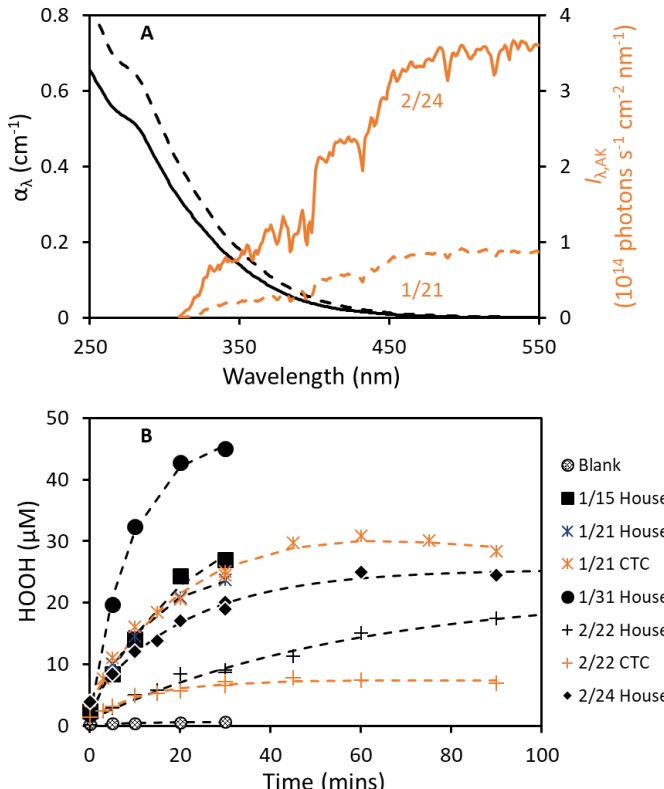

**Figure 1. (A) Absorbance spectra of two representative extracts (black lines) and their corresponding midday total (downwelling and upwelling) actinic fluxes in Fairbanks, Alaska (orange lines). Dashed and solid lines represent values for the composite samples of 1/21 and 2/24, respectively. (B) Examples of HOOH photoformation in PM extracts during solar simulator illumination in the lab. Rates for each extract, normalized to a standard photon flux ($j_{2NB} = 0.020\ s^{-1}$), are listed in Table S3.**

## 3.2 Photoformation rates and quantum yields of HOOH in PM Extracts

Figure 1a shows that there is significant overlap between light absorption in the samples and Fairbanks winter actinic fluxes, indicating there is potential for photochemistry in these particles. Indeed, we find that simulated sunlight illumination forms HOOH in each of the particle extracts. Typical time series of HOOH formation in extracts during irradiation are shown in Fig. 1b: the rates vary widely, but in all cases HOOH production is fastest initially and then slows and approaches (or reaches) a photostationary state at longer times, typically within 30 min. We observe some HOOH (0.1 – 5 μM) in our extracts prior to illumination, likely a result of formation in the dark during filter extraction, but there is no formation of HOOH in the dark samples during any experiment.



Photon-flux-normalized initial formation rates of HOOH in our experiments, i.e., $P_{HOOH,PME,EXP}$, range from 29 to 344 μM hr⁻¹ for the pH 1 extracts (Table S3). As shown in Figure 2A, the rates are typically 100 μM hr⁻¹ or lower except for the 1/31 sample, which was collected during the most polluted event of the field campaign and had a rate of 340 μM hr⁻¹. Although our extracts have PM mass/$H_2O$ mass ratios similar to cloud/fog drops (Table S1), our photoformation rates are much higher than values (0.04 - 3 μM hr⁻¹) previously reported for cloud water samples (Anastasio et al., 1994). The major reason for this

difference is that the Fairbanks PM extracts are much more light absorbing, possibly because they are fresher and less aged than the mostly summer cloud samples from the past work. In contrast, the quantum yields for the Fairbanks chromophores are similar to, or lower than, the values from the past cloud/fog waters, as discussed below. Finally, a portion of the faster rate for the Fairbanks PM extracts is because the standard lab photon flux in our current work is two times higher than the standard in the previous study, $j_{2NB} = 0.010$ s⁻¹ (Anastasio et al., 1994); we expect $P_{HOOH}$ to be proportional to photon flux.

The rate of HOOH formation in an extract is equal to the rate of light absorption times the quantum yield for HOOH formation ($\phi_{HOOH}$; i.e., the number of moles of HOOH formed per mole of photons absorbed). Figure 2B shows the rate of light absorption for each PM extract under our simulated sunlight conditions at a constant photon flux. These rates have the same temporal pattern as the rates of HOOH formation in Figure 2A, suggesting that the quantum yields are relatively constant across the field campaign. Figure 2C shows that this is generally true: values of $\phi_{HOOH}$ in the pH 1 House extracts range from 0.08% to

0.14%, with an average (± 1σ) of 0.11 (±0.02)%. The House extract at pH 4, with a quantum yield of 0.03%, is not included in this average since Figure 2C suggests that HOOH photoformation decreases with increasing pH, which we explore in the next section. While rates of HOOH formation in the four pH 1 CTC PM extracts are sometimes similar to the corresponding House values (Figure 2A), quantum yields for the CTC samples are all lower than the corresponding House results (Figure 2C and Table S3). This is likely because of differences in PM composition between the two sites. Comparing the two sets of

extracts, there is no clear difference in metal concentrations (Table S4), but concentrations of the major ions (e.g., $NH_4^+$, $SO_4^{2-}$, and $NO_3^-$) were often roughly twice as high in the CTC extracts (Table S2). In addition, while parameters that indicate chromophore abundance, such as $\alpha$ and DOC, were not significantly different between corresponding House and CTC extracts, AAE values were higher for the House site (Table S1). This difference in chromophore composition is consistent with results from Robinson et al. (2023), who found significant differences in $PM_{2.5}$ concentration and composition across Fairbanks during

ALPACA. In addition, the difference in PM size, with House samples being $PM_{2.5}$ and CTC samples being $PM_{0.7}$, likely contributes to the site difference in composition and quantum yields. Apparent HOOH quantum yields for our simulated sunlight experiments are at the lower end of values determined previously during 313 nm illumination of cloud water samples (0.02% – 1.1%; Anastasio et al., 1994) or an aqueous extract of particles from the high Canadian Arctic (0.75%; Anastasio and Jordan, 2004). Part of this difference is probably due to illumination wavelength, since $\phi_{HOOH}$ decreases with increasing

wavelength (Arakaki et al., 1995), but differences in sample composition probably drive most of the difference.



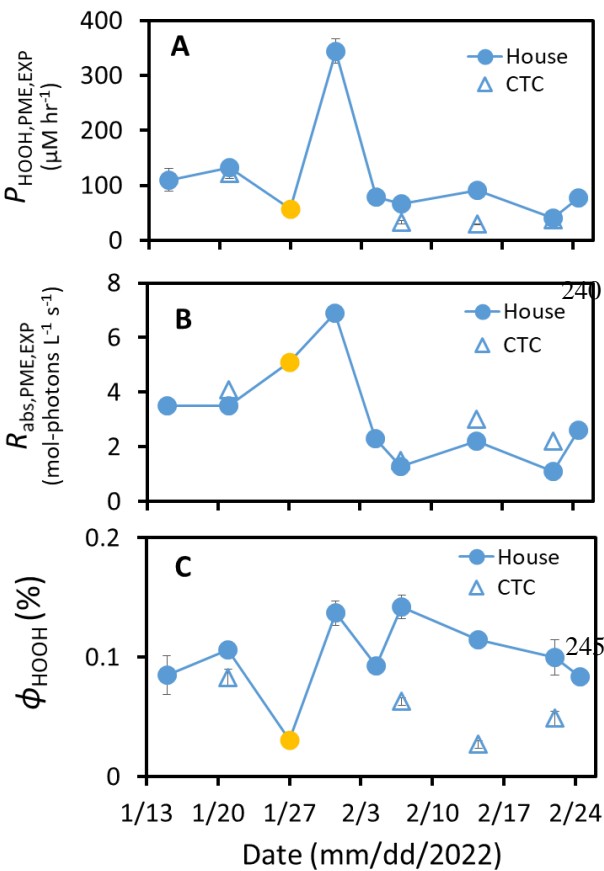

**Figure 2. (A) Experimentally measured photoformation rates of HOOH in aqueous extracts of PM collected from the House (filled circles) and CTC (open triangles) sites, normalized to a constant photon flux of $j_{2NB} = 0.020$ s$^{-1}$. Blue symbols are pH 1 extracts, while the gold symbol is pH 4. (B) Rate of light absorption in the extracts under our standard laboratory simulated sunlight. (C) Quantum yield of HOOH formation. Error bars for panels A and C are 1 standard error, determined from the kinetic fit of the experimental data (panel A) or propagated from this error (panel C). There are no errors shown in panel B, but these are expected to be smaller than the symbols, i.e., RSE < 5%.**



### 3.3 Rate constant for HOOH loss

In our HOOH experiments we also determined the first-order rate constant for hydrogen peroxide loss, $k_{HOOH,PME,EXP}$: values range from 1.2 to 6.5 hr$^{-1}$ (Table S3). Direct photodegradation of HOOH should be a minor sink in our experiments and the

inorganic S(IV) concentration should be negligible since there was HOOH in each extract at time zero. In contrast, metals are likely a major sink (Song et al., 2023; Zepp et al., 1992), although their oxidation states (and, therefore, reactivities with HOOH) are uncertain. We estimated the contribution of metals to $k_{HOOH,PME,EXP}$ based on three inputs: (1) measured concentrations of dissolved Fe, Cu, and Mn in each extract (Table S4), (2) assuming that approximately 80%, 5%, and 100%, respectively, of the three dissolved metals are in their reduced forms (i.e., Fe(II), Cu(I), and Mn(II)) during illumination

(Deguillaume et al., 2005; Siefert et al., 1998), and (3) rate constants of reaction with HOOH from the literature (Song et al., 2021; Watts et al., 2005). Based on these inputs, our calculated rate constant for HOOH loss due to metals ($k_{HOOH,PME,Metals}$) on average ($\pm 1\sigma$) accounts for 79 ($\pm$ 40)% of the measured value, indicating that transition metals are major sinks for HOOH in our illuminated extracts. The fraction of dissolved metal that is in its reduced form is significantly uncertain (Deguillaume et al., 2005; Mao et al., 2017; Siefert et al., 1998), which muddies our assessment of the contribution of metals to HOOH loss.

For example, if we assume that 90% and 10% of dissolved Fe and Cu, respectively, are in their reduced forms, then these metals account for an average of 105 ($\pm$ 60)% of $k_{HOOH,PME,EXP}$. Alternatively, if we use our base case Fe(II) and Cu(I) percentages (80% and 5%, respectively) but assume all dissolved Mn is present as Mn(III), then metals account for an average of 154% ($\pm$ 85%) of the HOOH rate constant for loss (Figure S2).




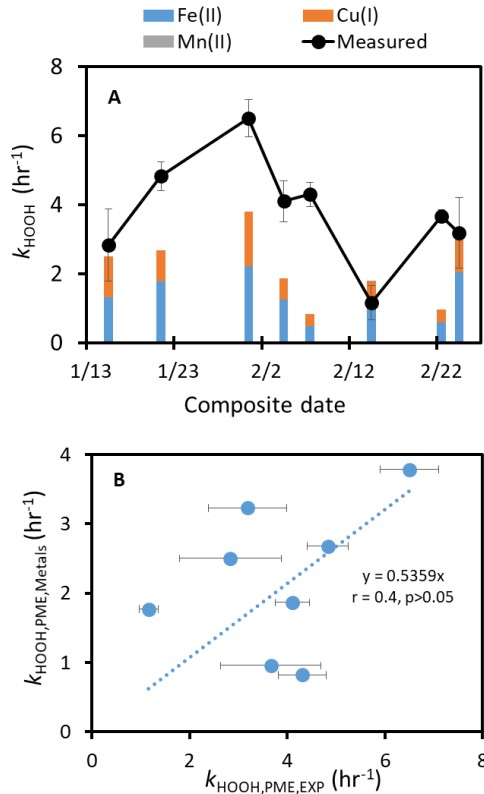

**Figure 3. Rate constants for the loss of HOOH, both experimentally determined ($k_{HOOH,PME,EXP}$) and calculated ($k_{HOOH,PME,Metals}$) based on reactions with dissolved Fe, Cu, and Mn in each extract. We calculated values of $k_{HOOH,PME,Metals}$ using: (1) measured dissolved Fe, Cu, and Mn concentrations in each extract (Table S4), (2) assuming 80%, 5%, and 100% of the dissolved iron, copper,**
**and manganese were present as Fe(II), Cu(I), and Mn(II), respectively (Deguillaume et al., 2005; Siefert et al., 1998), and (3) reaction rate constants of 70 M$^{-1}$ s$^{-1}$ for HOOH + Fe(II), 7000 M$^{-1}$ s$^{-1}$ for HOOH + Cu(I) (Song et al., 2021) and $4 \times 10^{-4}$ M$^{-1}$ s$^{-1}$ for HOOH + Mn(II) (estimated from data of Watts et al., 2005). The Mn(II) contribution to HOOH loss is too small to see in panel A; an upper bound for HOOH loss by manganese, assuming all manganese is present as Mn(III), is shown in Figure S4.**

**3.4 Effect of pH on light absorption, HOOH photoformation, and HOOH loss in PM extracts**

We next examined how optical properties, HOOH photoformation rates, and HOOH loss rate constants vary with pH. We focused on pH 1 and 5 since this range covers most of the acidities calculated for Fairbanks particles during the ALPACA campaign (Campbell et al., 2024). Figure S3A shows that the absorption spectra are very similar at pH 1 and 5. Previous studies have reported an increase in absorbance with increasing pH of aqueous particle extracts (Anastasio & Jordan, 2004;
Cai et al., 2018; Phillips et al., 2017; Teich et al., 2017), cloud water (Hennigan et al., 2023), and bulk solutions of





representative BrC compounds (Zhao et al., 2015). This is likely due to deprotonation of acidic chromophores in BrC to form more strongly absorbing species ( Lee et al., 2014; Phillips et al., 2017). The DOC values of the pH 1 and pH 5 particle extracts are the same (Table S1), so the MAC results mirror the absorbance results (Figure S3B). The ratio of absorbance at pH 5 compared to at pH 1 (Figure S3C) is close to 1.0 at 300 nm, reaches 1.1 near 365 nm, and peaks at 1.7 near 480 nm; however,

the absolute difference in absorbance at any wavelength is very small. The AAE decreases with pH, from 8.2 at pH 1 to 7.3 at pH 5 (Table S1).

As shown in Figure 4, the rate of HOOH photoformation significantly decreases with increasing pH. For the 1/21 CTC composite, the ratio of HOOH formation rates is 1 : 0.66 : 0.37 at pH 1, 3, and 5, respectively, while in the 2/22 CTC composite, which only has data at pH 1 and 5, the ratio of HOOH production is 1 : 0.12. There are at least two possible reasons why

HOOH formation is faster at lower pH. The first is that triplet excited states of aromatic carbonyls can be protonated in this range, resulting in a protonated triplet that usually reacts more quickly (compared to the unprotonated triplet) with phenols to generate HOOH (Anastasio et al., 1997; Ma et al., 2021; Smith et al., 2014). A second possible explanation is that the higher concentrations of transition metals at lower pH (Table S4) result in more photochemically active metal-organic complexes, which can produce HOOH (Song et al., 2023; Ye et al., 2021; Zuo and Hoigné, 1993).

The rate constants for HOOH loss also decrease with increasing pH (Figure 4). For the 1/21 CTC composite, the ratio of $k_{HOOH,PME,EXP}$ is 1 : 0.95 : 0.50 at pH 1, 3, and 5, respectively, while the pH 1 : pH 5 ratio in the 2/22 CTC composite is 1 : 0.22. While it is unclear whether transition metals play a role in HOOH formation in our samples, they dominate the HOOH sink and can explain its pH dependence. As shown in Figure S4, the decline in $k_{HOOH,PME,EXP}$ with increasing pH generally matches the reduction in the calculated rate constant for HOOH loss from metals, a consequence of decreasing dissolved metal

concentrations (Table S4); however, this relationship is complicated by the fact that the rate constants of metal-HOOH reactions can also depend on pH (e.g., Kremer, 2003).



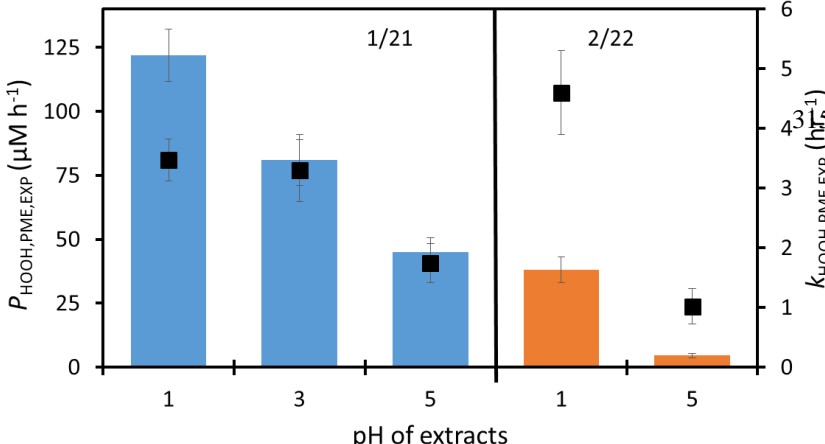


**Figure 4: Photoformation rates of HOOH (bars) and first-order rate constants for HOOH loss (square markers) as a function of extract pH. The left sample with blue bars is the CTC 1/21 composite, while the right sample with orange bars is the CTC 2/22 composite.**

### 3.5. Predicting HOOH photoformation rates in ambient particles

The rates of light absorption and HOOH formation that we discuss above (e.g., Figures 1b and 2a) are for PM extracts illuminated with simulated sunlight in the laboratory. To adjust these to rates expected in suspended particles during the ALPACA campaign, we need to make two conversions: (1) from laboratory photon flux to the midday Fairbanks sunlight of each composite and (2) from our dilute extracts to the concentrated conditions of aerosol liquid water (ALW). The first conversion, to Fairbanks sunlight conditions, reduces rates of light absorption in the dilute extracts by roughly a factor of five. As shown in Table S3 and Figure S1, the rate of light absorption under Fairbanks midday winter sunlight ($R_{abs,PME,AK}$) is, on average, 20 ($\pm$ 10)% of the rate under our standard laboratory condition ($R_{abs,PME,EXP}$). Accordingly, predicted HOOH formation rates in PM extracts under Fairbanks sunlight are also roughly 20% of the rates measured in the lab with simulated sunlight (Figure S1).

The second conversion, from dilute extracts to concentrated ALW conditions, requires that we understand how the production rate of HOOH formation varies with solute concentration. To explore this, we extracted squares from the 2/14 CTC composite filters with five different volumes of pH 1 sulfuric acid per filter square, from 0.30 to 10.0 mL (Table S5). We express the dilution of a given extract using concentration factor (CF, section 2.2), which is the inverse of the solvent volume used per square, so that higher CF values represent more concentrated extracts. As shown in Tables S5a-c and Figure S5, the sample parameters (e.g., rate of light absorption and DOC concentration) are all strong linear functions of CF, as expected for a dilution series. This shows that our extraction procedure effectively extracts water-soluble chromophores from the particles across the





entire range of dilution. It also indicates that rotary evaporation, which was used to make the two most concentrated extracts, did not introduce artifacts, consistent with our recent work (Ma et al., 2023).

We then illuminated each extract and measured [HOOH] to determine the production rate of HOOH and its rate constant for
345 loss as a function of dilution in the 2/14 composite. Figure 5 shows that $P_{HOOH}$ and $k_{HOOH}$ are both directly proportional to extract concentration, indicating that $P_{HOOH}$ is first order in chromophore concentration and $k_{HOOH}$ is first order with respect to metals and other sinks, at least over the concentration range of our dilution series.

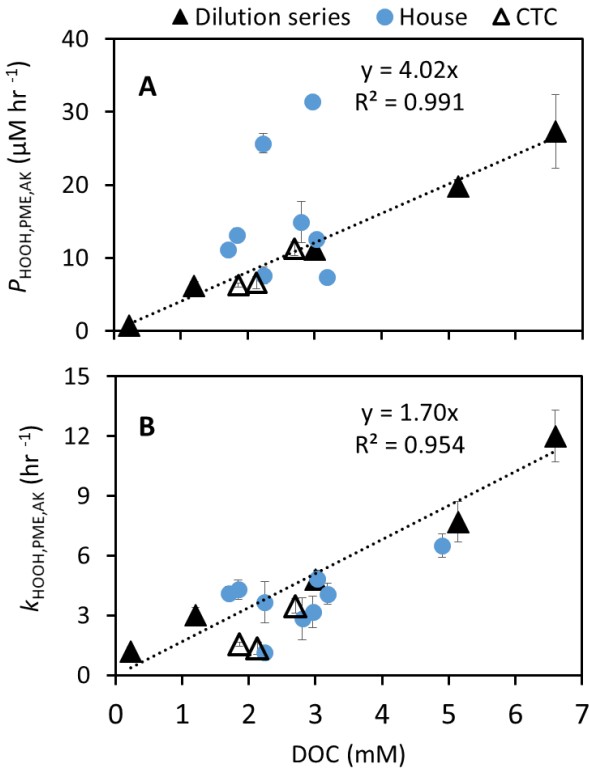

**Figure 5. Results from the dilution experiment for the 2/14 CTC pH 1 composite (filled triangles) shown with results from the other**
350 **pH 1 CTC (open triangles) and House (blue circles) composites. (A) Photoformation rate of HOOH as a function of DOC concentration (as a proxy for extract concentration). The 1/31 House sample is outside the plot boundaries, at 4.9 mM DOC and $P_{HOOH,PME,AK}$ = 71 μM hr$^{-1}$, and so is not shown. (B) Rate constant for loss of HOOH versus DOC. The dotted lines in the two panels represent linear regression fits (with _y_ intercepts fixed at zero) to just the dilution series data (filled triangles). Error bars on points represent 1 standard error from the kinetic fits of the experimental data.**

355



Based on the experimentally determined linear relationships of $P_{\text{HOOH,PME,AK}}$ and $k_{\text{HOOH,PME,AK}}$ with extract concentration (Figure 5), we extrapolated our extract measurements to aerosol liquid water conditions using

$$P_{\text{HOOH,ALW,AK}} = P_{\text{HOOH,PME,AK}} \times \frac{[\text{DOC}]_{\text{ALW}}}{[\text{DOC}]_{\text{PME}}} \qquad (5)$$

Here $[\text{DOC}]_{\text{PME}}$ represents the concentration of dissolved organic carbon measured in our PM extracts and $[\text{DOC}]_{\text{ALW}}$ is the corresponding ALW concentration, estimated from the extract DOC, modeled aerosol liquid water (Campbell et al., 2024), and the air volume sampled per filter square, as described in Heinlein et al. (In preparation). The [DOC] ratio on the right-hand side of equation 5 represents how much more concentrated solutes are in ALW compared to in the corresponding extract. Values range from 2300 to 39000 (Heinlein et al., In preparation). We use an analogous equation to determine the rate constant for HOOH loss in ALW. Values for the parameters used to extrapolate to ALW conditions are in Table S6.

The resulting dilute extract and concentrated ALW results are shown in Figure 6. Predicted rates of HOOH formation in ALW under Fairbanks sunlight are enormous, ranging from 0.04 to 0.5 M hr$^{-1}$ (Table S6). As best as we can tell, there is only one previous estimate of HOOH formation in particle water, 0.02 M h$^{-1}$, which was based on an extrapolation of measurements from an aqueous extract of ambient particles from the high Arctic in Canada (Anastasio and Jordan, 2004).

We also extrapolated the loss rate constant of HOOH from dilute extracts to ambient particle conditions (Figure 6 and Table S6). Calculated values in ALW from our extrapolation range from $(0.07-2) \times 10^5$ hr$^{-1}$, corresponding to HOOH lifetimes of 0.02 to 0.5 s, with metals as the primary sink (Figure 3). However, our extracts are missing inorganic S(IV), which should be a major sink for HOOH in Fairbanks particles given the high gas-phase SO$_2$ amounts (Moon et al., 2024; Simpson et al., 2024). While ambient particles from ALPACA contain significant amounts of both inorganic and organic S(IV) (Dingilian et al., 2024), these species decompose and/or evaporate in the time between filter collection and extract illumination. This is shown by the fact that in our illumination experiments, all extracts at time zero contained HOOH, which would have been destroyed if there had been residual inorganic S(IV). Heinlein et al. (In preparation) estimated the concentration of inorganic S(IV) in ALW for each composite extract by matching their modeled HOOH contribution to secondary sulfate to the contribution determined from sulfate isotope measurements (Moon et al., 2024). We use these inorganic S(IV) concentrations, along with kinetic components described in supplemental section S1, to calculate the rate constant for HOOH loss due to S(IV) in ALW, i.e., $k_{\text{HOOH,ALW,S(IV)}}$. Based on these values, we calculated the ALW sink for HOOH due to inorganic S(IV), which is missing from our extract extrapolations. As shown in Figure 6, the inorganic S(IV) sink for HOOH, with values in the range of $(200-10000) \times 10^5$ hr$^{-1}$, is orders of magnitude larger than the mostly transition-metal-mediated sink extrapolated from extracts. In contrast, Song et al. (2023) reported that Fe and Cu were the main sinks of aqueous-phase HOOH in winter particles from the North China Plain (NCP); this difference from our ALPACA result is at least partly due to much higher metal concentrations and somewhat lower SO$_2$ in the NCP study (Song et al., 2021).



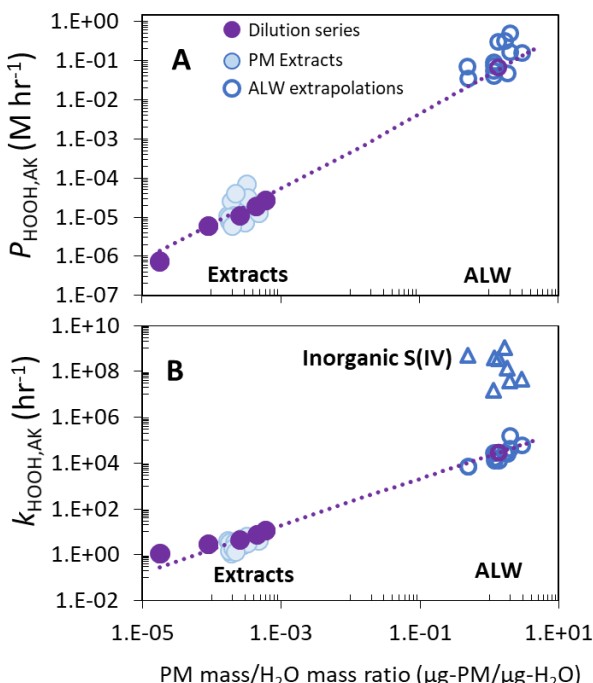

**Figure 6. (A) HOOH photoformation rates and (B) rate constants for HOOH loss under Fairbanks sunlight conditions as a function of solute concentration for the pH 1 samples. Filled light blue points above the "Extracts" label are House and CTC values measured in the lab in dilute PM extracts. Solid purple points represent lab results from the 2/14 CTC dilution series shown in Figure 5. The purple lines represent first-order extrapolations to the ALW condition for this sample, with the ALW results represented by the open purple circles. Open dark blue points above the "ALW" labels are estimated ALW values for the other House and CTC samples based on extrapolations from the lab results. In panel B, the dark blue open triangles represent the calculated rate constants for HOOH loss due to reaction with inorganic S(IV) in the particles, based on modelled S(IV) concentrations from Heinlein et al. (In preparation).**

### 3.6 Rate of sulfate production by in-particle HOOH

Since inorganic S(IV) is, by far, the largest sink for HOOH in ALW for our samples, essentially all the HOOH formed in particles will undergo reaction with S(IV) to form sulfate. Therefore, the rate of loss of S(IV) ($L_{S(IV),ALW,AK}$) due to reaction with HOOH is approximately equal to the rate of HOOH photoproduction, $P_{HOOH,ALW,AK}$. We convert this in-particle rate of S(IV) oxidation to an air-volume-normalized rate of sulfate formation, $P_{S(VI),Air,AK}$ (in units of μg m$^{-3}$-air hr$^{-1}$) using:

$$P_{S(VI),Air,AK} = L_{S(IV),ALW,AK} \times ALWC \times MW_{S(VI)} \times 10^6 \ , \qquad (6)$$



where $L_{S(IV),ALW,AK}$ is the rate of loss of S(IV) by HOOH in the ALW (mol S(IV) $L^{-1}$-$H_2O$ $hr^{-1}$), which we set equal to $P_{HOOH,ALW,AK}$ (mol HOOH $L^{-1}$-$H_2O$ $hr^{-1}$), ALWC is the aerosol liquid water content (L-$H_2O$ $m^{-3}$-air), $MW_{S(VI)}$ is the molar mass of sulfate (96 g $mol^{-1}$), and $10^6$ converts g to µg.

Figure 7A shows that the rate of sulfate formation from photoproduced HOOH in particles roughly mirrors the measured concentrations of particulate sulfate in Fairbanks. Based on isotopic data (Moon et al., 2024), an average of 62 (± 12)% of total sulfate during ALPACA is primary, while 38 (± 12)% is secondary. The dominant pathway for secondary sulfate is HOOH oxidation, which accounts for an average of 33 (± 14)% of total sulfate formed in the Fairbanks atmosphere (Moon et al., 2024). Our estimated rates of sulfate formation from BrC-photoproduced HOOH range from 0.05 – 0.5 µg $m^{-3}$ $hr^{-1}$, with an average (± 1σ) of 0.15 (± 0.14) µg $m^{-3}$ $hr^{-1}$. Using the composite-specific rates, we estimate the concentration of secondary sulfate that would be produced near midday over a period of two hours, which is a typical residence time for air during ALPACA (Cesler-Maloney et al., 2024). As shown in Figure 7B, the concentration of secondary S(VI) from HOOH ranges from 0.1 to 1.0 µg $m^{-3}$. While the isotope approach cannot distinguish between HOOH formed in the gas phase and particle phase, two pieces of evidence suggest that particle photochemistry is the dominant source: (1) our measured rates of HOOH in PM extracts are rapid enough to account for the HOOH-produced secondary sulfate determined by isotopic measurements, and (2) the high $NO_x$ levels during ALPACA should make the formation of gas-phase HOOH negligible.



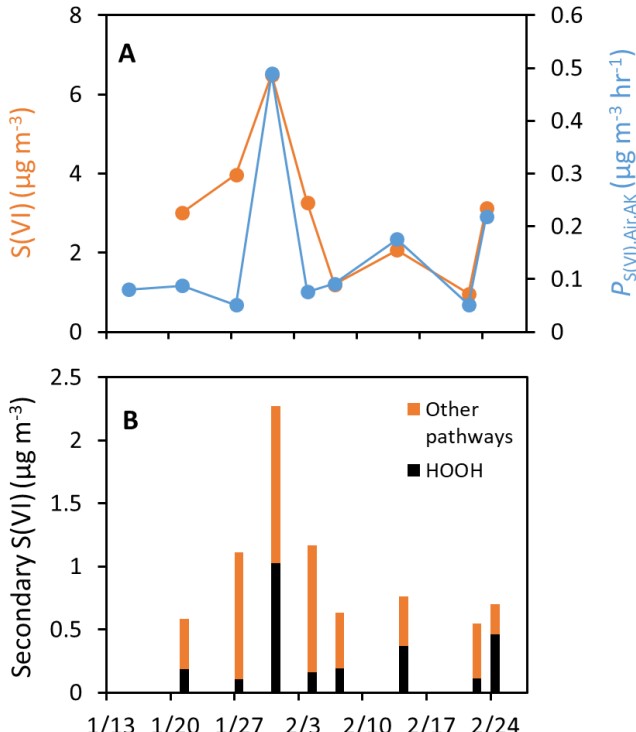


**Figure 7. (A) Concentrations of particulate sulfate (orange circles) and rates of sulfate formation by HOOH that was photochemically formed in particles (blue circles). (B) Concentrations of secondary sulfate produced by HOOH (black portion of bars) and other oxidation pathways (orange portion of bars). The amount of secondary sulfate formed by HOOH in a given composite was estimated by multiplying the rate of formation ($P_{S(VI),Air,AK}$) by 2.1 hr, a typical residence time for air during the campaign (Cesler-Maloney**
**et al., 2024). The secondary sulfate concentration from other pathways was determined as the difference between the total secondary sulfate (estimated as an average of the daily values from Moon et al. (2024) for a given composite) and the secondary sulfate from HOOH.**

## 4 Conclusions

We investigated the photoformation of hydrogen peroxide in aqueous extracts of wintertime particles collected from Fairbanks,
Alaska during the 2022 ALPACA campaign. While traditional wisdom suggests that photochemistry is negligible in high latitude regions during winter, all the extracts absorb significant amounts of sunlight and form HOOH at significant rates. We did not identify the chromophores responsible for this photochemistry, but brown carbon from biomass burning is likely a major source. Our extracts have solute concentrations that are typical of cloud/fog drops, roughly $10^3 - 10^4$ times more dilute than aerosol liquid water. To extrapolate our results to ALW conditions, we measured HOOH formation rates as a function of
dilution in extracts prepared from the same particle sample and found a linear relationship. Based on this, we estimate that rates of HOOH photoproduction in Fairbanks particles are on the order of 0.1 M hr⁻¹. Our calculations indicate that inorganic S(IV) is the major sink for hydrogen peroxide formed in particles, which means that the rate of sulfate production in ALW (in



mol L-aq$^{-1}$ hr$^{-1}$) is approximately equal to the rate of HOOH photoformation. This results in sulfate production rates from HOOH that are roughly 0.1 – 0.5 µg m$^{-3}$-air hr$^{-1}$. Based on this, particle-formed HOOH is a major contributor to secondary

sulfate in Fairbanks during ALPACA, in agreement with the isotope-derived result of Moon et al. (2024).

Our study demonstrates that HOOH is formed in particles even under the weak sunlight of a high-latitude site in winter. In part, this is because both the high albedo of snow cover and optical confinement of light within particles amplify the incident actinic flux (Heinlein et al., In preparation). In addition, the particles absorb significant amounts of sunlight, a consequence of abundant biomass burning and other sources of brown carbon during winter. In contrast to the rapid formation of HOOH in

particles, we expect negligible HOOH formation in the gas phase during the field campaign because HO$_2$$^\bullet$ concentrations will be suppressed by the high levels of NO. As described in Heinlein et al. (In preparation), illuminated particle extracts from ALPACA also form significant amounts of hydroxyl radical, triplet excited states of brown carbon, and singlet molecular oxygen. Together with our HOOH measurements, this indicates that particle photochemistry is significant even under the very cold and relatively low light conditions of Fairbanks during winter.


**Data Availability.** HOOH photoformation and spectral data are available upon request. ALPACA campaign data, including actinic fluxes, are available at https://arcticdata.io/catalog/portals/ALPACA.

**Author contributions.** MOS, LMDH, and CA developed the research goals and designed the experiments. MOS and LMDH with assistance from JH prepared the particle extracts, performed the photochemistry experiments, and measured the composite

extract compositions. AM, SK, TF, KE, BA, and MS collected the samples. FG and JHF collected and analyzed the Fairbanks actinic flux data. JD measured particle composition in the field. MOS and CA wrote the manuscript and revised it based on edits from co-authors.

**Competing interests.** At least one of the (co-)authors is a member of the editorial board of Atmospheric Chemistry and Physics

**Financial support.** We gratefully acknowledge funding from the National Science Foundation (Grant numbers 2109011 (UC

Davis), 2109023 (UNH), 2109098 (UH), and 1654104 (UC Irvine)) and the National Oceanic and Atmospheric Administration (Grant number NA20OAR4310295 to UW).

**Acknowledgements.** We thank Kerri Steenwerth and Kyle T. Sherbine (USDA) for assistance with ion analysis, Austin M. Cole (UC Davis) for analyzing metals, and Bill Simpson (UAF), Jochen Stutz and Jonas Kuhn (UCLA), and Thorsten Bartels-Rausch (PSI) for stimulating discussions.



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
