# Peer review of "Hydrogen Peroxide Photoformation in Particulate Matter and its Contribution to S(IV) Oxidation During Winter in Fairbanks, Alaska"

_EGUsphere, 2024_

## Author Comment (AC3)

**Response to Reviewers Comments**

"Hydrogen Peroxide Photoformation in Particulate Matter and its Contribution to S(IV) Oxidation During Winter in Fairbanks, Alaska" by Sunday et al.

Each reviewer comment is listed in italics and our response, in plain text, is directly below it.

**Anonymous Referee #1 Received and Published: 25 Nov 2024**

**Overall comments:**

*The authors studied HOOH photo-formation in aerosol extracts collected in Fairbanks, Alaska. The photo-formation rates are very fast because they used smaller amounts of acidic solution to extract chemicals in the aerosols, i.e., larger absorptivity over the 310–550 nm range. The fast HOOH photo-formation is likely to significantly contribute to S(IV) oxidation, even in the aerosols of high-latitude atmospheres. The authors also calculated the quantum yields of HOOH photo-formation to be about 0.001 at pH 1, a typical aerosol pH. This information is quite valuable and useful for estimating HOOH photo-formation in aerosols. The reviewer believes that the results presented in the manuscript are interesting and add valuable information to the environmental and atmospheric chemistry communities, likely stimulating further studies. Thus, the reviewer recommends the publication of this manuscript after the authors address the minor comments listed below in the revised manuscript.*

**Detailed comment:**

*Page 4, Line 103-104: It is mentioned that Milli-Q water was used for extraction and the extract was used to determine ion composition. Were the metal ion concentrations determined using the Milli-Q extract? Metal ion concentrations may be higher for the pH 1 extract compared to the Milli-Q extract.*

**Response:** We thank the reviewer for this comment. The metals were determined in both pH 1 and Milli-Q extracts, although we didn't show the Milli-Q results in the previous draft. All calculations of rate constants of HOOH loss due to metals (Section 3.3 and Fig. S2) used metal concentrations in pH 1 extracts. For comparison, we have now included the metal concentrations in Milli-Q extracts as Table S4b of this revised draft. We have revised the manuscript to indicate this point in the main text and caption of Table S4. As predicted by the reviewer, the metal concentrations are higher in the pH 1 extracts compared to the Milli-Q extracts.

*Page 13, Line 305-311: The loss rates of HOOH are discussed, and the trend shows slower rates as pH increases. If Fe(II) is a major sink for HOOH, the rates should increase as Fe(OH)+ (a favorable chemical species as pH increases) has faster reaction rate constants compared to Fe2+ (a major species at lower pH). The reviewer assumes that only inorganic Fe(II) is considered. In the aerosol extracts, it may be possible that, since DOC is much higher in the*

*aerosol extracts, a large fraction of Fe (II) may be bound to organics, and the Fe(II)-organic complexes may have similar reaction rate constants to that of Fe2+ and HOOH.*

**Response:** We thank the reviewer for this insightful comment. Multiple factors - including the concentration and speciation of Fe(II) - can influence the loss rate constant of HOOH due to Fe reactions. The same is true for copper and manganese. While we find that the soluble Fe concentration in the extracts decreases with increasing pH (Table S4a), we do not know its speciation or whether it changes with pH. We agree that Fe(II)-organic complexes could be the major form of reduced iron, while $Fe(OH)^+$ and $Fe^{2+}$ could also be significant, depending on pH. The complexity of our samples and our lack of knowledge about Fe(II) speciation makes it difficult to provide further clarity. Therefore, we assumed that the metals are present as their hexa-aqua forms (e.g., $Fe^{2+}$) in our calculations. We have added more information about these issues, and the uncertainty in metal speciation, in the main text.

**Anonymous Referee #2 Received and Published: 11 Dec 2024**

*General comments:*

*This paper presents a HOOH formation pathway via illuminating particle extracts, and analyzes its influence factors such as light absorption properties, transition metals, acidity. Then, the authors estimated HOOH formation rates in ambient particles, and confirmed its dominant role in sulfate production. Altogether, the results are meaningful and the paper is clear. I have a number of points that I feel should be addressed.*

Specific comments:

1. *The manuscript is generally well-written, but certain sections can benefit from additional clarification, particularly regarding the $H_2O_2$ formation mechanisms involved in the photochemical aging of actual $PM_{2.5}$ in the introduction (Liu et al., ES&T, 2021, doi.org/10.1021/acs.est.1c04453; Zhang et al., ES&T, 2020, doi.org/10.1021/acs.est.0c01532).*

   **Response:** We thank the reviewer for this comment and pointing us to additional references. The work of Liu et al. (2021) proposes a scheme where HOOH is formed on the surface of dry particles (at low RH) and transported to the gas phase. The work of Zhang et al. proposed in-particle formation of HOOH and OH by water-soluble and water-insoluble matter and the contribution of these oxidants to $SO_2$ oxidation and photochemical aging. Both studies demonstrate in-particle formation of HOOH, which aligns with our hypothesis, and we have added them to our introduction (line 56).

2. *Line 266-268: Figure 3 should be noted somewhere in this sentence. In addition, there may be a mistake in the caption of Figure 3, i. e., the last sentence: ...is shown in Figure S2, not Figure S4.*

**Response:** We thank the reviewer for these observations. As noted by the reviewer, we have cited Figure 3A (line 273) accordingly, and revised the caption of Figure 3 to correctly cite Figure S2 (line 293).

3. *Line 297-304: As hydrogen peroxide is a weak acid, I am not sure whether it can be formed via the displacement by strong acid (i. e., sulfuric acid) especially at lower pH. The authors can add some discussion as necessary.*

   **Response:** Thank you for your comment. We are not suggesting that HOOH is produced via acid displacement; we agree that this is insignificant in our samples since the p$K_a$ for HOOH is 11.6. Rather, our first proposed mechanism is triplet excited states of brown carbon reacting with phenols to produce HO$_2^{\bullet}$ radicals, which are then converted to HOOH by either HO$_2^{\bullet}$ (i.e., dismutation), phenols, or reduced transition metals. Both light-absorbing brown carbon compounds and phenols are ubiquitous in biomass burning particles. We discussed this mechanism in line 61- 64 of the introduction (Anastasio et al. 1997) and now expand on it some here. Our second proposed mechanism is photochemical reactions of metal-organic complexes, which would also be more important at low pH values because of increased soluble metal concentrations.

4. *"Heinlein et al. (in preparation)" was mentioned many times throughout the manuscript, which is not as the solid evidence for your conclusion. If the authors think it is important, the relevant data and results should be added in this study.*

   **Response:** Thank you for your comment. We submitted the draft of Heinlein et al. to the journal along with our manuscript, so we believe the reviewer has been able to see this in preparation work. We hope that the reviewer has seen that our current manuscript and that of Heinlein et al. are both connected (e.g., they use the same samples) but also very different (e.g., Heinlein et al. focuses on measurements of hydroxyl radical, triplet excited states, and singlet molecular oxygen). Because the two manuscripts have different first authors, we have tried to give credit appropriately throughout both works, which has resulted in numerous cross-citations.

5. *Although I strongly agree with the conclusion that the dominant pathway for secondary sulfate is HOOH oxidation, especially near midday, the authors should compare their findings with actual observational data from previously published literatures. This comparison would enhance the credibility of their results.*

   **Response:** We thank the reviewer for this comment. As discussed in section 3.6, we compare our results with those of Moon et al (2024), who conducted sulfate isotope analysis on the same CTC PM samples to assess which oxidants drive secondary sulfate formation. We see generally good agreement with their results on the importance of HOOH as an oxidant, which gives us confidence in our measurements. In addition, the range of sulfate formation rates by in-particle formation of HOOH in our study (0.05 – 0.5 µg m$^{-3}$ hr$^{-1}$) agrees well with rates (0.11 ± 0.15 µg m$^{-3}$ hr$^{-1}$) determined by Song et al (2021) for wintertime particles collected on the North China Plain. Interestingly, although the rates of HOOH formation in these two studies are similar, S(IV) oxidation in

the Song et al. study was dominated by transition metal ions. We have added this information to the revised text.

6. *The source of the precursors (e.g., BrC) of HOOH was considered to be mainly from biomass burning or residential wood combustion, but the detailed discussion was lacking in this paper. I suggest the authors had better add them.*

**Response:** Previous studies have found that organic PM from biomass burning accounts for up to 52% of PM mass in Fairbanks (Ward et al., 2012; Wang and Hopke, 2014; Kotchenruther, 2016; Ye and Wang, 2020; Haque et al., 2021). In addition, a recent study conducted on Fairbanks particles during the 2022 ALPACA campaign showed that biomass burning organic aerosol (BBOA) contributed $28 \pm 18\%$ of total organic aerosol (OA) in Fairbanks' particles as determined by aerosol mass spectrometry (Ijaz et al., 2024). We have added the above discussion to the revised manuscript (line 188 -193) to show that BrC from biomass burning is an important component of organic aerosol that can act as a photosensitizer driving photochemical reactions, including HOOH formation.

References

Haque, M. M., Kawamura, K., Deshmukh, D. K., Kunwar, B., and Kim, Y.: Biomass burning is an important source of organic aerosols in interior Alaska, Journal of Geophysical Research: Atmospheres, 126, e2021JD034586, 2021.

Ijaz, A., Temime-Roussel, B., Chazeau, B., Albertin, S., Arnold, S. R., Barrett, B., Bekki, S., Brett, N., Cesler-Maloney, M., Dieudonne, E., Dingilian, K. K., Fochesatto, J. G., Mao, J., Moon, A., Savarino, J., Simpson, W., Weber, R. J., Law, K. S., and D'Anna, B.: Complementary aerosol mass spectrometry elucidates sources of wintertime sub-micron particle pollution in Fairbanks, Alaska, during ALPACA 2022 , EGUsphere [preprint], https://doi.org/10.5194/egusphere-2024-3789, 2024.

Kotchenruther, R. A.: Source apportionment of PM2. 5 at multiple Northwest US sites: Assessing regional winter wood smoke impacts from residential wood combustion, Atmospheric Environment, 142, 210-219, 2016.

Moon, A., Jongebloed, U., Dingilian, K. K., Schauer, A. J., Chan, Y.-C., Cesler-Maloney, M., Simpson, W. R., Weber, R. J., Tsiang, L., Yazbeck, F., Zhai, S., Wedum, A., Turner, A. J., Albertin, S., Bekki, S., Savarino, J., Gribanov, K., Pratt, K. A., Costa, E. J., Anastasio, C., Sunday, M. O., Heinlein, L. M. D., Mao, J., and Alexander, B.: Primary Sulfate Is the Dominant Source of Particulate Sulfate during Winter in Fairbanks, Alaska, ACS EST Air, 1, 139–149, https://doi.org/10.1021/acsestair.3c00023, 2024

Song, H., Lu, K., Ye, C., Dong, H., Li, S., Chen, S., Wu, Z., Zheng, M., Zeng, L., Hu, M., and Zhang, Y.: A comprehensive observation-based multiphase chemical model analysis of sulfur

dioxide oxidations in both summer and winter, Atmospheric Chemistry and Physics, 21, 13713–13727, https://doi.org/10.5194/acp-21-13713-2021, 2021

Wang, Y. and Hopke, P. K.: Is Alaska truly the great escape from air pollution?-long term source apportionment of fine particulate matter in Fairbanks, Alaska, Aerosol and Air Quality Research, 14, 1875-1882, 2014.

Ward, T., Trost, B., Conner, J., Flanagan, J., and Jayanty, R.: Source apportionment of PM2. 5 in a subarctic airshed-Fairbanks, Alaska, Aerosol and Air Quality Research, 12, 536-543, 2012.

Ye, L. and Wang, Y.: Long-term air quality study in Fairbanks, Alaska: Air pollutant temporal variations, correlations, and PM2. 5 source apportionment, Atmosphere, 11, 1203, 2020.